# Conducting Violence and Mental Health Research with Female Sex Workers during the COVID-19 Pandemic: Ethical Considerations, Challenges, and Lessons Learned from the Maisha Fiti Study in Nairobi, Kenya

**DOI:** 10.3390/ijerph20115925

**Published:** 2023-05-23

**Authors:** Mary Kung’u, Rhoda Kabuti, Hellen Babu, Chrispo Nyamweya, Monica Okumu, Anne Mahero, Zaina Jama, Polly Ngurukiri, Emily Nyariki, Mamtuti Panneh, Pooja Shah, Alicja Beksinska, Erastus Irungu, Wendy Adhiambo, Peter Muthoga, Rupert Kaul, Helen A. Weiss, Janet Seeley, Joshua Kimani, Tara S. Beattie

**Affiliations:** 1Partners for Health and Development in Africa, Nairobi P.O. Box 3737-00506, Kenya; 2Global Health and Development, London School of Hygiene and Tropical Medicine, London WC1H 9SH, UK; 3Department of Medicine, University of Toronto, Toronto, ON M5S1A8, Canada; 4MRC International Statistics and Epidemiology Group, Department for Infectious Disease Epidemiology, London School of Hygiene and Tropical Medicine, London WC1E 7HT, UK

**Keywords:** violence, mental health, COVID-19, female sex workers, ethics

## Abstract

Conducting violence and mental health research during the COVID-19 pandemic with vulnerable groups such as female sex workers (FSWs) required care to ensure that participants and the research team were not harmed. Potential risks and harm avoidance needed to be considered as well as ensuring data reliability. In March 2020, COVID-19 restrictions were imposed in Kenya during follow-up data collection for the Maisha Fiti study (*n* = 1003); hence data collection was paused. In June 2020, the study clinic was re-opened after consultations with violence and mental health experts and the FSW community. Between June 2020 and January 2021, data were collected in person and remotely following ethical procedures. A total of 885/1003 (88.2%) FSWs participated in the follow-up behavioural–biological survey and 47/47 (100%) participated in the qualitative in-depth interviews. A total of 26/885 (2.9%) quantitative surveys and 3/47 (6.4%) qualitative interviews were conducted remotely. Researching sensitive topics like sex work, violence, and mental health must guarantee study participants’ safety and privacy. Collecting data at the height of COVID-19 was crucial in understanding the relationships between the COVID-19 pandemic, violence against women, and mental health. Relationships established with study participants during the baseline survey—before the pandemic—enabled us to complete data collection. In this paper, we discuss key issues involved in undertaking violence and mental health research with a vulnerable population such as FSWs during a pandemic. Lessons learned could be useful to others researching sensitive topics such as violence and mental health with vulnerable populations.

## 1. Introduction

The first case of COVID-19 in Kenya was announced on 13 March 2020 [1]. The non-pharmaceutical interventions initiated by the government to prevent the spread of the infection included cessation of all movement by road, rail, or air in and out of the Nairobi metropolitan area, Kilifi, Kwale, and Mombasa counties; a nationwide curfew from 7:00 p.m. to 5:00 a.m.; closure of all bars, hotels, and places of worship; wearing masks while in public areas; frequent hand washing and sanitising; and keeping a physical distance of 1.5 m from other people. For female sex workers (FSWs) working in Nairobi county, these restrictions presented challenges that disrupted their lives and livelihoods [2,3]. Night curfews led to reduced working hours, closure of venues where they met clients, and fewer clients wanting to buy sex. The Ministry of Health (MoH) recommendation of physical distancing and putting on face masks complicated sex work, which involves close contact with clients during negotiations and sexual intercourse [2,4]. The cessation of movement to and from Nairobi County meant FSWs could no longer move around to solicit clients elsewhere in the country, reducing clients and income potential.

Sex workers also suffered isolation from their peers and other contacts, forcing some to adopt alternative ways to find clients. These changes included a shift to online platforms for those with smartphones, while others solicited clients during the day, before curfew, which negatively affected their volume of clients [2]. The Kenyan Government provided no financial support or other social protection measures to help alleviate the loss of income for FSWs, which resulted in economic hardship. In addition, during the restrictions, there were increased reports of intimate partner violence (IPV) and sexual and gender-based violence among all women and girls (i.e., not only FSWs) in Kenya; a “pandemic within a pandemic” [5]. Among sex workers, a spike in violence was explained by increased attacks by clients and violence from the police and community members who blamed FSWs for spreading COVID-19 [3]. According to Delor and Hubert, “vulnerability is the relational, contextual, and the process aspect of risk” [6]: it is the product of exposure to risk, the capacity to deal with the risk, and the risk of suffering the consequences of the crisis exposure. FSWs can be considered a vulnerable population due to the various individual, social/cultural, and structural factors characterising their lives and occupation [7,8]. Under normal circumstances, low income, unemployment, economic stress, emotional insecurity, and social isolation are risk factors that can increase the occurrence of IPV, non-partner violence (e.g., from clients or the police), and mental health problems; COVID-19 restrictions exacerbated these risk factors [9].

The experiences of FSWs in Nairobi were also reported elsewhere. Globally, women who sell sex are frequently vulnerable due to their marginalised status in society and their poor socio-economic status [2,10,11]. During the COVID-19 pandemic, the global network of sex workers reported that sex workers experienced hardship, loss of income, and increased discrimination and harassment due to the criminalisation of sex work [12,13]. Despite governments providing different forms of economic relief to economically vulnerable populations, as sex work is criminalised in most countries, it was difficult for FSWs to access this support [8,13]. In addition, sex work is not recognized as work, and the sex industry is not recognised as a form of small business ownership. Thus, sex workers were also unable to access labour protection or economic support that would otherwise aid small businesses [8,11].

Collecting data at the height of the COVID-19 disruptions to FSW’s daily lives had the potential to provide insights into how the increased hardship many participants faced affected their experiences of violence, harmful drinking, and mental health, as well as changes to cortisol levels, inflammation, and HIV risk—factors which our study, “Maisha Fiti”, was in the process of investigating [14,15]. However, researching violence and mental health problems among vulnerable populations, such as FSWs, requires careful consideration of the potential risks, research participants’ harm avoidance/minimisation, and data reliability [14,16,17]. Researchers should only collect data when it is safe to do so while prioritising the safety and support of participants [18]. Research team members require training to collect these data and to be equipped with information on support services for violence survivors when needed [14,18]. The WHO and United Nations guide on collecting violence data during COVID-19 stressed the importance of protecting women’s confidentiality and minimising harm or distress to both participants and the research team [14]. The guidance also recommended that direct questions on experiences of violence should not be included within population-based rapid assessments because of the risks posed to the victims of IPV if conversations were overheard [18]. This was particularly important during the lockdown as many women had to spend long periods of time at home with their abusers [18,19]. Similarly, guidelines on mental health research during the COVID-19 pandemic suggested that research should be prioritised based on its significance to the current situation; the benefits to participants should outweigh the risks [20,21,22].

The Maisha Fiti study was a longitudinal study that aimed to understand the impact of upstream risk factors, such as the experience of violence and mental health problems, on systemic and genital immunology. As such, research included collecting data on sensitive topics, including recent experiences of violence, mental health problems, and suicidal behaviour. This paper aims to discuss the specific ethical considerations and the methodological challenges we encountered, and the strategies which we applied during the follow-up data collection phase of the Maisha Fiti study to ensure the successful completion of this study research while protecting the safety and well-being of participants and study staff.

## 2. Materials and Methods

### 2.1. Study Setting and Design

The Maisha Fiti study was a mixed methods longitudinal study with 1003 FSWs in Nairobi, Kenya. The study aimed to: (i) assess the prevalence, severity, and frequency of FSW’s experience of violence, mental health morbidity, and problematic alcohol/substance use, and how these differ by HIV status; (ii) define the impact of these factors on genital inflammation; and (iii) examine which structural and behavioural factors—particular to FSW populations—impact on antiretroviral (ARV) uptake and adherence, and whether non-adherence to ARV is associated with increased drug resistance and HIV disease progression.

During the administration of the behavioural-biological survey questionnaire, participants were asked detailed questions about their experiences of recent emotional, physical, sexual, and financial violence by intimate partners, clients, police, and others, using the validated WHO violence against women and girls survey tool [23]. Participants were also asked about their current drinking and substance use risk (WHO ASSIST) and their mental health status (depression (PHQ-9), anxiety (GAD-7), post-traumatic stress disorder (HTQ-17), suicidal thoughts), using validated tools. We collected urine, blood, and genital samples to test for HIV, STIs, pregnancy, systemic, and genital inflammation, as well as hair samples to test for cortisol levels. The study team included a specialist counsellor who was situated in the study clinic and was available for immediate and free referral for any participant needing support.

Before the baseline data collection (June–December 2019), the study team received three weeks of intensive training. Clear referral pathways were created for FSWs reporting violence, poor mental health, or suicidal behaviours. Anyone requiring additional support was referred to external specialists for expert review and management. The research team held weekly structured debriefing sessions for the study staff, with support from the study counsellor and the study principal investigators, to support their mental well-being. There was free outsourced psychological counselling support for staff members who needed further confidential support and expert management. Baseline behavioural-biological quantitative and qualitative data were collected in June–December 2019, just before the COVID-19 pandemic. Follow-up data collection started in January 2020 and was one-third completed when the study clinic was closed in March 2020 due to the pandemic. From an ethical and research perspective, we were keen to complete follow-up data collection if this could be done safely, as it would allow for study completion.

### 2.2. Study Participants and Consultations on Managing COVID-19 Safe Research

At the start of the study, from a sampling frame of all women aged 18–45 years (*n* = 10,292) who had attended one of seven FSW SWOP (Sex Workers Outreach Program) clinics across Nairobi in the preceding 12 months, 1200 were randomly sampled, with 1039 eligible, and 1003 (96.5% participation rate) enrolled in phase I of the quantitative survey [24]. Of these, 40 participants were randomly selected to also participate in the baseline qualitative interviews. When the follow-up bio-behavioural survey commenced in January 2020, we had planned to complete 1003 quantitative survey interviews and 47 qualitative interviews by June 2020. As noted above, the facility was temporarily closed due to COVID-19 restrictions between March and June 2020. Before the closure, we had completed 365/1003 (36.3%) follow-up interviews.

During the closure, the study co-investigators, research team, and FSW community leaders stayed in touch and planned for the safe resumption of the study activities. Several consultations were held between the sex worker community leaders, sex workers hired as outreach workers, and the research team before finalising plans to re-open the research facility, as per Kenya Ministry of Health guidelines. Discussions were held with the Sex Workers Research Advisory Committee (SWARC) and the Community Advisory Board (CAB) members to obtain their buy-in and support before the study data collection was resumed. These consultations helped ensure the meaningful involvement of all key stakeholders while being fair and respectful to those involved or affected by the study closure [25].

Key considerations during the consultations were threefold. (i) Should the study resume? Was this something the FSW community supported, or would they prefer the study remain closed? (ii) Should the study move to remote interviews and biological sample collection? (iii) Should the study clinic re-open and in-person interviews be conducted?

On balance and following inputs from external experts as well as FSW representatives, we decided that guaranteeing the safety and well-being of participants concerning violence, mental health, and suicidal behaviour meant that conducting the interviews in-person within the confidential clinic space, and with direct access to the study counsellor, was the preferred option, even though this could increase the COVID-19 risk for participants and the research team. For the tiny minority of participants who had relocated overseas or to rural homes, it was agreed that the study team would consider, on a case-by-case basis, conducting remote interviews by mobile phone. Guidelines and study team training were developed and delivered to support remote data collection that ensured participants’ confidentiality and safety (regarding partner violence and suicidal behaviours). Participants who were interviewed remotely had their informed consent and data recorded using a digital device, and the interviewer made a written record.

### 2.3. Ethical Considerations

At baseline, participants provided written informed consent before the survey interview, during which they were informed we would have a follow up behavioural-biological survey 6–12 months later. Before the follow-up interview, participants again provided informed consent. This was either written (in-person interviews) or verbal and audio-recorded (remote telephone interviews). As part of the informed consent process, potential risks and risk management strategies were discussed with participants, and participants were able to have any questions they had answered before consenting to participate. All participants were reimbursed 500 KSH (5 USD) at each survey visit to cover their time and travel costs.

Changes to the Maisha Fiti study protocol were approved by the Kenyatta National Hospital—University of Nairobi Ethics Review Committee and Research Ethics Committees at the London School of Hygiene & Tropical Medicine and the University of Toronto. A joint decision was made to re-open the research facility on 8 June 2020, following the Kenyan MOH COVID-19 containment guidelines and approval by these various ethics boards.

## 3. Results

The phase III follow-up survey was conducted from June 2020 to March 2021 and reached 885/1003 (88.2%) participants for the behavioural–biological survey and 47/47 participants for the qualitative in-depth interviews. Of these, a small minority 26/885 (2.9%) of quantitative surveys and 3/47 (6.4%) of qualitative interviews were conducted remotely by telephone. This was possible as we had completed our baseline interviews before the pandemic and thus established relationships and trust with study participants; if this had not been the case, we would not have proceeded with data collection during the pandemic. Factors associated with loss-to-follow-up at endline were HIV status (higher follow-up among women living with HIV; *p* = 0.001) and reported recent violence from non-IPs (higher follow-up among those not reporting violence; *p* = 0.05) [24]. We also collected feedback from participants at the end of the questionnaire. We asked study participants two questions that were open ended: (i) How did answering the questions in this second round of the Maisha Fiti study make you feel? (ii) How did participating in the overall Maisha Fiti study make you feel? Of the 885 follow up participants, most (98%) reported positive reactions to the interviews. This included participants answering they felt fine, good, comfortable, happy, cared for, and relieved to have someone to talk to amid the COVID-19 crisis. In addition, some appreciated the clinical tests and treatments. A small minority (2%) felt that we should be supporting them financially during that period and not asking them research questions during this difficult time.

### 3.1. Modified in-Person Interviews

Standard operating procedures were developed to ensure adherence to the COVID-19 MOH prevention and control protocols to maintain staff and participants’ safety concerning COVID-19 risk at the facility premises. Activities under the guidelines summarised in Figure 1 included continuous sensitisation of the research team about COVID-19 symptoms, how it is spread, and prevention. Personal Protective Equipment, including surgical masks and face shields, were provided and were obligatory. All team members were urged to observe the MOH guidelines on infection prevention [26]. The call script used to mobilise study participants before COVID-19 was re-designed to additionally screen for COVID-19. If anyone reported any symptoms, they were requested to stay home until these were resolved. The participants’ facility visits were scheduled to minimise crowding at the reception, and research processes were fast-tracked to reduce the time spent at the research facility.

According to the MOH COVID-19 containment measures, temperature screening and hand sanitising were applied to all individuals entering public buildings. Only two people were allowed in the elevator at any given time in the building housing the study facility. Study participants had to follow the same rules while coming to the study facility. Hand washing points were provided, with all study participants encouraged to comply. A repeat temperature check was taken at the facility reception, and a new surgical mask was issued to everyone with a request to wear it properly throughout their stay. A contact log was kept of everyone who visited the facility for easy contact tracing in case of a COVID-19 infection, where names, phone numbers, and temperature were recorded. The contact tracing data were kept safe and separate from the study data to protect participants’ confidentiality. The reception area was also restructured, and seats were clearly labeled to enhance physical distance (see Figure 2). The facility was frequently disinfected throughout the day, paying attention to the most-touched surfaces like doorknobs, seats, surfaces, and the washroom keys (see Figure 3). Enough supplies (facemasks, sanitisers, soap) were available to facilitate these COVID-19 containment measures. The interviews and biological sample collection were carried out in ventilated rooms (see Figure 4). To support participants who were unable/uncomfortable visiting the research facility in the central business district and those who feared using public transport, the research team also conducted outreach visits to the seven Sex Workers Outreach Program (SWOP) clinic facilities where participants could attend for their follow-up study visit if they preferred. COVID-19 significantly decreased earnings from sex work; hence, some women lacked the initial bus fare to travel from their homes to visit the research facility. The travel costs were reimbursed after women participated in the study, 500 KSH (~5 USD) at each visit. For participants who were willing to visit the study clinic from their rural home, additional bus fare reimbursements were provided based on the average fares charged by public transport companies for that route.

#### 3.1.1. Implementation of Telephone Interviews

Where telephone interviews were offered, these interviews were guided by new study guidelines (Figure 1), developed and aligned to ethics and best practices from other studies on violence in Malawi and Zimbabwe [27]. During the initial phone call with a participant, their interest was confirmed, and safety was established for the remote interview. The study procedures were then explained to those interested, and a time and day they would be comfortable taking part in the interviews were confirmed and scheduled accordingly. If the participant had concerns about their safety on the agreed day, the interview was rescheduled. These concerns included having their partners or family members around or present. Technological challenges were also assessed and participants were advised to charge their phones for their interviews, consider phone audibility, locate points in their house or compound where the signal was strongest, and ensure other people could not eavesdrop on their conversations. Safe words were agreed, such as: “the weather is sunny,” to terminate the interview when the participant wanted to indicate that they needed to end the phone call quickly. Reimbursement was sent via a mobile money transfer platform at the end of the interview (500 KSH (5 USD)). Tele-medicine and tele-counselling were also encouraged when a participant had particular health or mental health concerns. Feedback from FSWs during our consultations suggested that FSWs are used to managing their phones and their whereabouts in a way that many non-FSW women may not be. Thus, we found that the women involved in the telephone interviews were adept at managing the safe conduct of calls. In a few instances, the interviewer ended the call upon hearing the safe word, and the participant reached out again when it was safe for them to resume the interview.

#### 3.1.2. Challenges and Lessons Learned while Conducting in-Person Interviews

Challenges while conducting in-person interviews during the COVID-19 pandemic and how we navigated these are summarised in Table 1. One of the most common problems we faced was ensuring physical distancing. Sometimes participants came to the clinic in groups, making physical distancing difficult. This was resolved by scheduling appointments ahead of time and participants being urged to stop bringing their friends to the facility. Some FSWs did not want to leave sex work early to participate in the research study because the curfew hours meant they needed to do sex work during the day. A staff member would explain why they needed to come for the study without coercing them, although the fear of contracting COVID-19 was a major concern.

Some participants avoided the facility altogether since we were taking temperatures routinely. At the height of COVID-19 disruptions, anyone with a fever of greater than 38 °C was supposed to be screened for COVID-19 infection, and MOH quarantined those found positive at ksh.2000 ($16.63) per person per day (costs incurred by the infected person). At the same time, the curfew enforcement agents also quarantined anyone breaching the law at government-designated centres at their own expense. These strict rules were later eased, with the Kenyan government covering these expenses to encourage people to test for the virus [2]. Due to economic hardship, most of the FSWs had no telephone airtime and some phones were off; the study team had to reach several affected people using the home locator information provided at baseline via the sex worker outreach team.

Some participants’ expectations also shifted due to COVID-19, where healthcare came second to food and monetary assistance. The Maisha Fiti study team had challenges explaining to participants that their financial needs were beyond the resources of the study. This perceived inadequate support resulted in some participants not coming for follow-up visits as they felt let down, although we did refer them to partner organisations offering financial support.

#### 3.1.3. Challenges and Lessons Learned while Conducting Phone Interviews

The challenges we faced with the telephone interviews are summarised in Table 1. In addition to the challenges of finding a private place for the call for those who lived with children, partners, or other family members, there were also issues when telephones ran out of charge in the middle of the interview. In other cases, the network signal was deficient, making conversation difficult. In such situations, the interviews were rescheduled. There were cases of missed interviews, despite participants confirming they would be available. Poor concentration was also noticed, especially among participants with crying toddlers, making communication challenging. Such interviews were rescheduled to days and times when participants were free from family duties.

## 4. Discussion

We have described the challenges experienced during a longitudinal study with a key population in Nairobi, Kenya, which was disrupted partway through due to the COVID-19 pandemic, and the ethical and safety considerations we needed to consider before resumption. The Maisha Fiti study team had drawn its participants from SWOP clinics, where a trusting relationship with sex workers had been developed over the preceding decade. Indeed, an institution’s reputation with potential participants has been shown elsewhere to influence research participation [28]. Building on relationships created during the baseline interviews enabled us to proceed with the follow-up study data collection. Relationships rooted in trust and respect between researchers and study participants were crucial in ensuring study success, especially during extenuating circumstances such as a pandemic. Trust is essential to building and maintaining mutually respectful relationships, especially partnerships involving patients or community stakeholders and researchers, where there is often an inherent imbalance of power [29]. Building rapport with participants can also reduce response bias by motivating respondents to engage more deeply with the interview and give thoughtful, honest responses [16,30].

Although in-person visits for the study were encouraged, remote methods, supported by tele-counselling, helped reach participants who would have otherwise been missed. Other studies in Brazil, Britain, Nepal, Uganda, and Zimbabwe, which also collected follow-up data on violence against women and children (but not with FSWs), made similar decisions to include remote data collection methods during the COVID-19 pandemic as described [18]. Limiting remote interviews to only those in exceptional circumstances helped reduce potential violence and mental health risks to participants. Creating a COVID-19-safe facility environment enabled staff and participants to feel safe regarding COVID-19 infection risk by adhering strictly to MoH COVID-19 prevention and control guidelines.

## 5. Conclusions

Researching sensitive topics like sex work, violence, and mental health must guarantee study participants’ safety and privacy [17]. Collecting data at the height of the COVID-19 disruptions to FSW’s daily lives was crucial in understanding the relationships between the COVID-19 pandemic, violence against women, and mental health. Relationships we had established with participants before the pandemic, FSWs prior experiences of managing their phones, the prioritisation of in-person interviews, and the carefully planned strategies for conducting remote interviews enabled us to complete data collection without (to our knowledge) causing any harm. If these relationships had not been established during the baseline surveys, we would not have proceeded with data collection during the COVID-19 pandemic. Lessons learned from this study could be of use to others researching sensitive topics like violence and mental health with vulnerable populations in the future.

## Figures and Tables

**Figure 1 ijerph-20-05925-f001:**
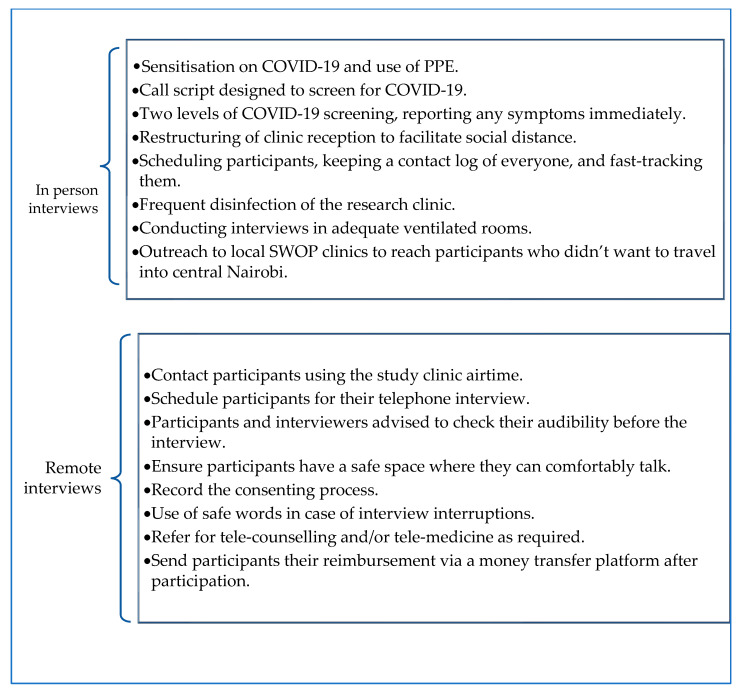
Safety measures for conducting face-to-face and remote interviews.

**Figure 2 ijerph-20-05925-f002:**
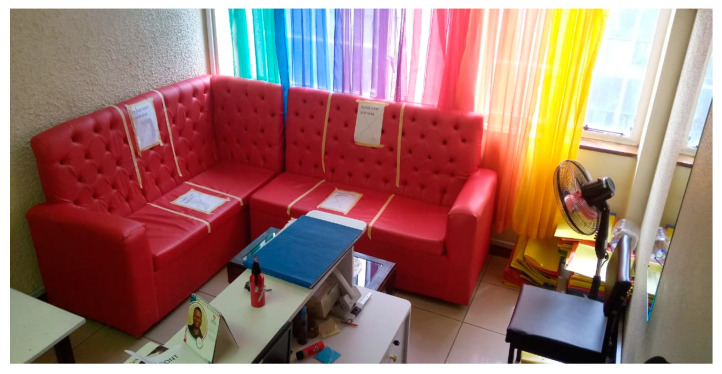
Restructured reception.

**Figure 3 ijerph-20-05925-f003:**
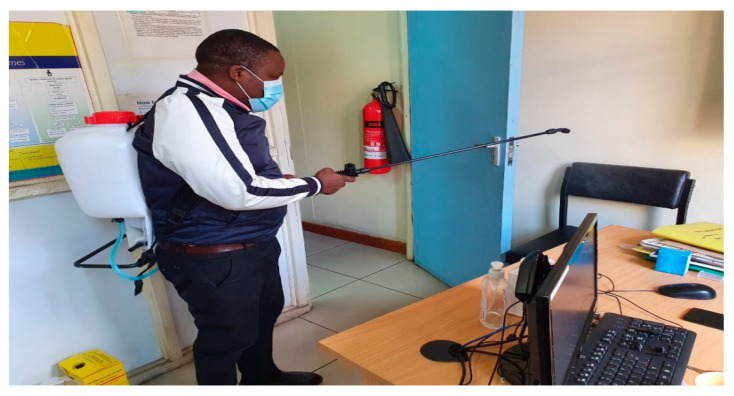
Fumigation of the facility.

**Figure 4 ijerph-20-05925-f004:**
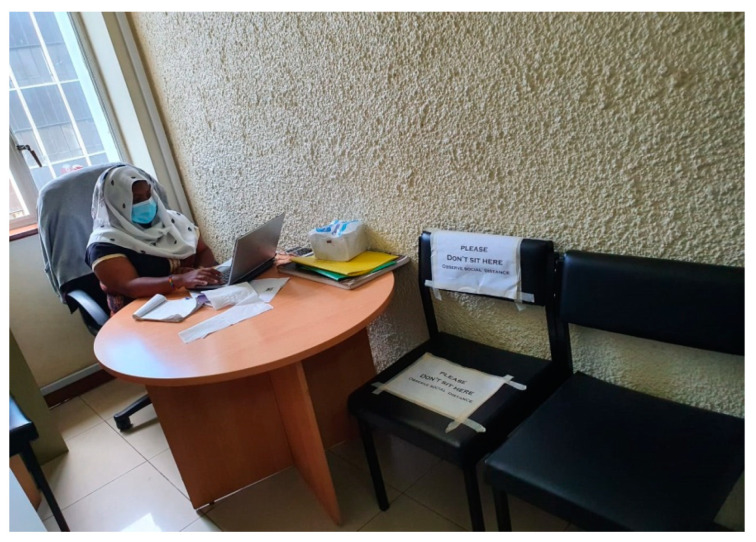
Safety measures in the rooms.

**Table 1 ijerph-20-05925-t001:** Unforeseen challenges and mitigation strategies while conducting follow-up interviews with FSWs in Nairobi, Kenya, during the COVID-19 pandemic.

Unforeseen Challenges	Mitigation Strategies
(i) Conducting face-to-face interviews	
▪ Fear of COVID-19—participants did not want to come to the facility, especially those aged 40+ years	▪ Assuring participants’ safety before their visit
▪ Participants show up in groups	▪ Scheduling participants per day and time
▪ Fear of being taken to COVID-19 quarantine facilities.	▪ Prescreening COVID-19 symptoms by telephone before scheduling the participant
▪ Half of the staff caught COVID-19	▪ Self-isolation and their contacts tested
▪ High expectations from the facility in terms of food and money when the participants came to participate	▪ Linking to organisations that could help the sex workers with food and other support
▪ Cessation of movement; hence participants were not able to come to Nairobi to participate physically	▪ We had to wait for the roads to be re-opened. Scheduling of some participants for remote interviews
▪ Participants lacked transport	▪ Reimbursement for bus tickets presented
▪ Some participants became homeless, hence affecting participation	▪ We waited for them to come when they felt able to participate (once their homelessness had been resolved)
▪ Time limitation since participants were working during the day and the facility was also operating during the day	▪ Fast-tracking such participants so that they can go to work after the interview
▪ Disinterested participants who blocked us or avoided Maisha Fiti’s calls	▪ Emphasising that participation is voluntary and honoring their right to withdraw from the study if they chose to
(ii) Conducting remote interviews	
Participants living with family, children or partners who could be violent	Use of safe words in case of interruptions
Participant’s phone receiving other call(s) during the interview	Waiting for participants to complete their call and return to the interview
Participants not honoring appointments	Rescheduling them
Poor network and inaudible phones	Participants urged to find a space with good network coverage

## Data Availability

The data that support the findings of this study will be available on request from the corresponding author from June 2023 (two years after study data collection was completed). The data are not publicly available due to privacy or ethical restrictions.

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
