# Peer review of "Conducting Violence and Mental Health Research with Female Sex Workers during the COVID-19 Pandemic: Ethical Considerations, Challenges, and Lessons Learned from the Maisha Fiti Study in Nairobi, Kenya"

_ijerph, 2023, doi:10.3390/ijerph20115925_

Round 1

Reviewer 1 Report

This manuscript details some important issues when conducting research with vulnerable people under conditions of increased vulnerability. In many respects, this submission makes for a good METHODS paper.

INTRODUCTION: This section is well-written and provides a clear rationale for the study. The focus on ethics over 'findings' makes for a refreshing perspective. The description of systemic impacts of COVID on the country as a whole, and the target population in particular, reflected some nuanced issues with this group.

METHOD: Sample was clearly defined and recruitment procedure was clear. Consultations with relevant stakeholders (e.g., SWARC and CAB) were good to see in this section and reflected an inclusive approach on the part of the researchers. 

RESULTS: Given the nature of data-collection (i.e., quantitative survey) and the focus on the approach, the results are really points of discussion. While a descriptive approach is OK for this element of the inquiry, it would have been of benefit to have seen a more rigorous analytical strategy. In some ways, this ms forms a type of discussion paper - rather than a study per se - of the range of issues involved in executing this research. Table 1 and 2 could have been presented in a less cluttered way (i.e., no gray-tones and maybe group the items in a more cohesive way).

DISCUSSION/CONCLUSIONS: This section was adequate. The insights from this work would be valuable for other research of this type (irrespective of the status of the sample). In this sense, the pragmatic approach discussed by the authors makes for a useful addition to the literature, especially given the people and conditions in which it was conducted.

GENERAL: I may have missed this, but 'vulnerability' wasn't well-defined and risks being an over-inclusive term. As such, I would recommend altering the title to better reflect that this study was about research with 'female sex workers' - a sub-population who are prone to multiple vulnerabilities perhaps, but not defined as such. To be fair, the issues of vulnerability are evident in the introduction, but 'vulnerable populations' - especially during the Covid pandemic - would also suggest the elderly, those with poor access to medical/health services (vaccines, etc.), and those with pre-existing medical compromises, but not necessarily social vulnerabilities. Lastly, the authors efforts to humanise their participants is evident by attention to safety, wellbeing and approaches to minimising distress as part of the research journey.

Reviewer 2 Report

This paper considers the second phase of data collection for a study conducted during the Covid-19 pandemic about female sex workers and the safety and socio-economic risks they face. Several key features of the original study were presented and details. The significance of the study overall was set out. The main sections of the article discussed ethical considerations about re-launching and completing the second phase of the longitudinal data collection. Several challenges and sensitivities surrounding this were considered. In the main the article focuses on the safety aspects of relaunching and completing the fieldwork in terms of managing Covid-19 risks as well as risks to safety for the participants due to the risks they may potentially face in taking in part in social research. The rationales for decisions made, the involvement of sex worker organisations and a frontline clinic as a key research study site, and the involvement of ethics committee approvals - were considered. In several ways the article provides a detailed example of how and why a research team during the pandemic negotiated the return to in-person fieldwork in this case allowing also for several adaptions and flexibilities to promote participation in the research on a remote or adapted basis if needed. The safety of participants was evidenced as well considered. The adaptions undertaken to the data collection echoes several studies published about undertaking qualitative research during the pandemic although there are few studies published about research in this national setting and related to the sensitivity of the focus of this study. 

I would have been interested in a wider discussions and details about research ethics related to this study and the adaptions in the second phase of data collection completed during the pandemic. The key features of the ethical approach and requirements adopted for the study overall and the first phase of the study - would be worth knowing more about. This helps the reader to have a better understanding of the adaptions undertaken later and how the importance of ethical principles per se informed the second phase adaptions. For example, more information about how the participants for the study were identified and recruited to take part would be useful. More information about how the process of informed consent was designed and undertaken would be useful too. For example, was there going to be a second phase of informed consent from the outset for the second phase of data collection? Were participants given any incentives to take part in the study by the research team or frontline clinic, and what were the rationales for this? Were there any issues in terms of recruiting participants? I would also be interested to know what the position of the researchers' funding body (the medical research council) on what research can be conducted in-person and remotely during the pandemic, and how this influenced decisions made about the second phase of data collection. More information also about the second process of informed consent in the pandemic context would be useful. How were risks / risk management communicated with and discussed with participants as part of the informed consent process? 

Another area that would have been interesting to hear more about would be some more reflections or an assessment by the research team about the implications of the adaptions / relaunch for changes to the participant sample and its key characteristics, and the quantity or quality of the data collected. The authors here may want to point the reader to other publications for more detailed discussions but further brief discussion would be useful to help the reader to consider the adaptions made in a broader way. 

The article needs proof reading to amend a few minor language issues. 

Reviewer 3 Report

Dear Authors,

The issues of violence against women are essential! In the particular case of national policies that allow considering the profession of sex as regulated or permitted, this topic is highly relevant. The authors should contextualize the profession of sex in a global scale, following the COVID19 impact. However, the restrictions taken concerning COVID-19 are well known and identified, as other infections that have had a significant impact on a global scale, namely HIV. I believe this topic may be outdated and no longer attracts the reader's attention, in the pathway as the authors present it in the manuscript. Compared to other regulated professional activities where women can have a prominent place, I suggest reorganizing the study .

Round 2

Reviewer 2 Report

The authors have addressed considerations raised in the review process. The diagrams have been altered and improved. The article needs a final thorough proof read and edit.